# Performance of a postnatal metabolic gestational age algorithm: a retrospective validation study among ethnic subgroups in Canada

Steven Hawken,[1,2,3] Robin Ducharme,[3] Malia S Q Murphy,[1] Katherine M Atkinson,[1,4] Beth K Potter,[1,3,2] Pranesh Chakraborty,[5,6] Kumanan Wilson[1,3,2,7]

[1]Clinical Epidemiology Program, Ottawa Hospital Research Institute, Ottawa, Ontario, Canada
[2]School of Epidemiology, Public Health and Preventative Medicine, University of Ottawa, Ottawa, Ontario, Canada
[3]uOttawa, Institute for Clinical Evaluative Sciences, Ottawa, Ontario, Canada
[4]Department of Public Health Sciences, Karolinska Institute, Stockholm, Sweden
[5]Department of Paediatrics, University of Ottawa, Ottawa, Ontario, Canada
[6]Newborn Screening Ontario, Children's Hospital of Eastern Ontario, Ottawa, Ontario, Canada
[7]Department of Medicine, University of Ottawa, Ottawa, Ontario, Canada

**Correspondence to**
Dr Kumanan Wilson;
kwilson@ohri.ca

## ABSTRACT

**Objectives** Biological modelling of routinely collected newborn screening data has emerged as a novel method for deriving postnatal gestational age estimates. Validation of published models has previously been limited to cohorts largely consisting of infants of white Caucasian ethnicity. In this study, we sought to determine the validity of a published gestational age estimation algorithm among recent immigrants to Canada, where maternal landed immigrant status was used as a surrogate measure of infant ethnicity.

**Design** We conducted a retrospective validation study in infants born in Ontario between April 2009 and September 2011.

**Setting** Provincial data from Ontario, Canada were obtained from the Institute for Clinical Evaluative Sciences.

**Participants** The dataset included 230 034 infants born to non-landed immigrants and 70 098 infants born to immigrant mothers. The five most common countries of maternal origin were India (n=10 038), China (n=7468), Pakistan (n=5824), The Philippines (n=5441) and Vietnam (n=1408). Maternal country of origin was obtained from Citizenship and Immigration Canada's Landed Immigrant Database.

**Primary and secondary outcome measures** Performance of a postnatal gestational age algorithm was evaluated across non-immigrant and immigrant populations.

**Results** Root mean squared error (RMSE) of 1.05 weeks was observed for infants born to non-immigrant mothers, whereas RMSE ranged from 0.98 to 1.15 weeks among infants born to immigrant mothers. Area under the receiver operating characteristic curve for distinguishing term versus preterm infants (≥37 vs <37 weeks gestational age or >34 vs ≤34 weeks gestational age) was 0.958 and 0.986, respectively, in the non-immigrant subgroup and ranged from 0.927 to 0.964 and 0.966 to 0.99 in the immigrant subgroups.

**Conclusions** Algorithms for postnatal determination of gestational age may be further refined by development and validation of region or ethnicity-specific models. However, our results provide reassurance that an algorithm developed from Ontario-born infant cohorts performs well across a range of ethnicities and maternal countries of origin without modification.

### Strengths and limitations of this study

► This validation study has successfully demonstrated that a gestational age estimation algorithm performs well across infants from diverse backgrounds without modification.
► Population-based design: The validation cohort included 300 132 live-born infants born between April 2009 and September 2011 who underwent newborn screening at Newborn Screening Ontario.
► Defining ethnic subpopulations: Landed immigrant status, rather than self-reported ethnicity, was used as a surrogate measure for identification of ethnic subpopulations.
► Model development: The model demonstrates poorer performance among more severely preterm infants, in part due to smaller numbers of preterm infants available for model development.

## INTRODUCTION

Knowledge of gestational age at the time of birth is vital for ensuring adequate provision of newborn care and for assessing population-level estimates of the burden of preterm birth to guide allocation of health services resources and targeted global health initiatives.[1 2] In jurisdictions with challenging socio-economic conditions and/or limited access to antenatal care and ultrasound dating technology due to rurality, determination of gestational age can be challenging. Other antenatal dating methods, including last menstrual period and fundal height measurements, are hampered by poor recall history and a high prevalence of low birth-weightl. Where prenatal estimations are unavailable or unreliable, a variety of standardised fetal assessments have been developed for clinicians seeking to determine fetal maturation after birth. Commonly used postnatal assessments that score infants on neurological and physical criteria are only accurate to within

3–4 weeks of true, ultrasound-validated gestational age.[3] World health and philanthropic organisations are now seeking novel ways of determining gestational age at the time of birth, both to improve individual care and to provide reliable, high-quality data for population surveillance.

Secondary analysis of newborn screening samples routinely collected within the first few days of an infant's life has emerged as a unique opportunity for postnatal gestational age assessment. We and others have recently demonstrated the accuracy of postnatal gestational age algorithms derived from newborn screening data in three independent North American cohorts.[4–6] A significant limitation of the approaches published to date has been the predominance of white infants in the populations used for model validation. Metabolic profiles are subject to biological variation as a result of *in utero* environmental exposure,[7] and recent work suggests that ethnic diversity within a population needs to be considered when establishing newborn screening reference intervals for some conditions.[8] This study sought to validate a postnatal gestational age metabolic model in ethnic subpopulations in Ontario, Canada.

## MATERIALS AND METHODS
### Study design
We conducted a population-based retrospective validation study of infants born in Ontario, Canada using a combination of linked health administrative, newborn screening and immigration datasets maintained by the Institute for Clinical Evaluative Sciences. The Newborn Screening Ontario (NSO) database includes the analyte profiles of each infant completing newborn screening in the province (>99% of all infants born in Ontario). Over 40 screening analytes and analyte ratios including acylcarnitines, amino acids, endocrine markers, enzymes and coenzyme markers among others are available in the NSO database (table 1). Maternal country of origin was used as a surrogate marker of infant ethnicity and was ascertained from Citizenship and Immigration Canada's

Landed Immigrant Database. This study was approved by the Ottawa Health Science Network Research Ethics Board, Ottawa, Canada (20140724-01H).

### Original model
The postnatal gestational age estimation model previously published by our group included 249 700 infants born between April 2007 and March 2009. The details of the original model have been described previously.[9] Briefly, infants who were identified as positive for any disorder screened for by NSO were excluded, as were infants with unsatisfactory samples, and missing gestational age and birth weight. Because complete metabolite profiles were required to score new observations, infants with missing analyte data or other covariates including gestational age, were excluded from this validation exercise. The infants excluded due to missing covariates constituted <5% of the cohort. Gestational age was based on best obstetrical estimate (last menstrual period, dating ultrasound or a combination). It is to be noted that >99% of women in Ontario receive at least one ultrasound during the course of pregnancy.[10]

The data were randomly partitioned into model development (50%), validation (25%) and test (25%) subsamples. Multiple linear regression was performed in the model development set, in which all analyte main effects were included in the model, as well as birth weight and sex. For analytes and birth weight, squared and cubic terms were also included. A stepwise variable selection algorithm was then conducted and all pairwise interactions were considered for inclusion. The Schwartz Bayesian Criterion, which rewards improved model fit and penalises model complexity, was used in variable selection. Once this process was complete, the mean square error (MSE) for the fitted model at each step was calculated in the independent validation sample subset, and the model with the smallest MSE was selected. This approach provides a high level of protection from overfitting in the final model.[9] Parameter estimates for the fitted model were fixed, and used to score (ie, estimate gestational age) in the independent test sample subset (n=62 434) and model performance characteristics (r-square, MSE, proportion with observed vs estimated gestational age within 1 week) were calculated. The final regression model included a total of 311 parameter estimates, including main effects, quadratic and cubic effects plus interaction terms.

The deviation of each calculated gestational age from the true gestational age of each infant is the residual model error for that infant (in unit, weeks). The residual model error can be positive or negative depending on the direction of the difference. The MSE is the mean of each of those residual errors after squaring it (also rendering all values positive) for all infants. MSE is presented in the units of weeks.[2] Taking the square root of the MSE yielded an overall 'average deviation' in unit, weeks.

Gestational age was then dichotomised to distinguish term or preterm infant subgroups. Model performance

**Table 1** Newborn screening analytes used in model development

| | |
|---|---|
| Acylcarnitines | C0, C2, C3, C3DC, C4, C4DC, C4OH, C5, C5:1, C5OH, C5DC, C6, C6DC, C8, C8:1, C10, C10:1, C12, C12:1, C14, C14:1, C14:2, C16, C16OH, C16:1OH, C18, C18:1, C18:2, C18OH, C18:1OH |
| Amino acids | Alanine, arginine, citrulline, glycine, leucine, methionine, ornithine, phenylalanine, tyrosine, valine |
| Endocrine markers | 17-Hydroxyprogesterone, thyroid-stimulating hormone |
| Enzyme and coenzyme markers | Galactose-1 phosphate uridylyltransferase, biotinidase |

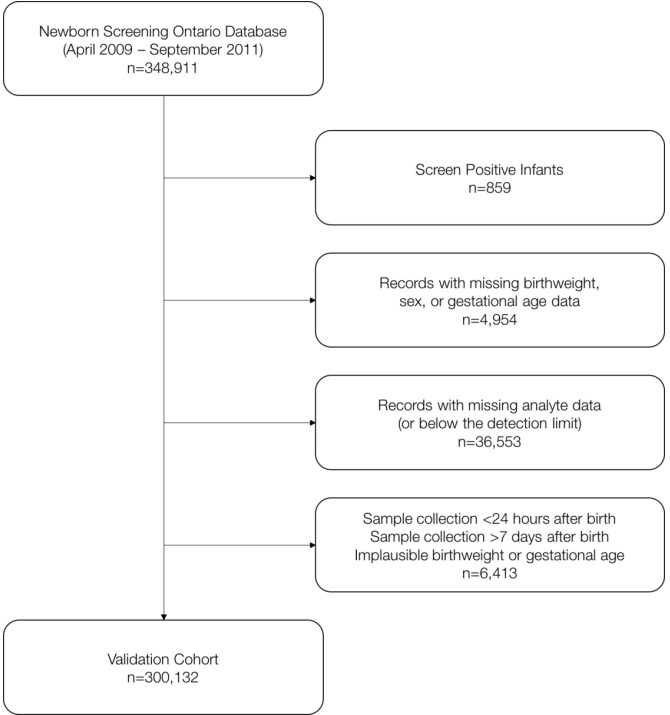

Newborn Screening Ontario Database
(April 2009 – September 2011)
n=348,911

Screen Positive Infants
n=859

Records with missing birthweight,
sex, or gestational age data
n=4,954

Records with missing analyte data
(or below the detection limit)
n=36,553

Sample collection <24 hours after birth
Sample collection >7 days after birth
Implausible birthweight or gestational age
n=6,413

Validation Cohort
n=300,132

**Figure 1** Cohort creation strategy.

to correctly classify infants across two thresholds of preterm birth categories by logistic regression analysis was assessed (area under the receiver operating characteristic curve (AUC), sensitivity, specificity and positive predictive value). Preterm birth thresholds were <37 weeks gestational age, the clinical threshold for preterm birth, and ≤34 weeks gestational age, a threshold that represents the lower limit of the late preterm period. Infants born ≤34 weeks gestational age (early preterm and severe preterm infants) are at increased health risk compared with those born late preterm or at term. Estimated gestational age identified by multiple linear regression was used as a continuous independent variable in logistic regressions to determine the probability of preterm birth.[9]

### Validation cohort
The validation cohort included 300 132 live-born infants born between April 2009 and September 2011 who underwent newborn screening at NSO. A total of 348 098 records were available for the prescribed study period. Records with missing or implausible (data entry error) data on birth weight, sex and gestational age were removed from the analysis. Infants who received a 'positive' flag for any one of the newborn screening conditions were excluded. Also excluded were those with missing newborn screening analyte values, or those for which the samples were collected before 24 hours and after 7 days of birth. Figure 1 summarises the logic used to create the study cohort. The validation cohort was independent from the data used in the development of the original model.

### Model validation
The coefficients from the reference linear regression model were fixed, and used to score the validation cohort. Calculated gestational age was used as the independent variable in logistic regression models to estimate the dichotomous categories of preterm birth, for which model performance characteristics including AUC, sensitivity, specificity and positive predictive value were calculated. Confidence Intervals for AUC were also calculated, using the approach described previously by Hanley and McNeil.[11] SAS V.9.4 was used for all statistical analyses.

### RESULTS
The validation dataset included 230 034 infants born to non-immigrant mothers and 70 098 born to immigrant mothers. The five most common countries of maternal birth were India (n=10 038), China (n=7468), Pakistan (n=5824), The Philippines (n=5441) and Vietnam (n=1408). The most common countries from the African continent were Somalia (n=833) and Nigeria (n=800), respectively. Descriptive characteristics of the validation cohort are provided in table 2.

Overall model performance characteristics for non-immigrant populations and the top eight landed immigrant subgroups are presented in table 3. Model performance characteristics from the original validation cohort are provided for comparison. Absolute gestational age estimation was within 1 week in continuous linear regression models; root mean squared error ranged from 0.98 weeks (maternal Indian heritage) to 1.15 weeks (maternal Somalian heritage).

Model performance among preterm infants is provided in table 4. Among non-immigrants, our algorithm performed comparably to our original validation with AUC for classifying infants as term versus preterm; ≥37 vs <37 weeks gestational age, AUC=0.958 and >34 vs ≤34 weeks gestational age, AUC=0.986.

Our gestational age estimation model was able to discriminate between dichotomous preterm birth categories of immigrant infants with robust precision. For discrimination of ≥37 vs <37 weeks gestation, AUC ranged from a 0.927 among infants of maternal Somalian descent to 0.964 among infants of maternal Nigerian heritage. Similarly, the model was able to discriminate well between >34 weeks and ≤34 weeks gestational age, with AUC ranging from 0.966 for Nigerian and Bangladeshi infants to 0.994 among Filipino infants.

### DISCUSSION
This study validates a postnatal gestational age estimation model in subgroups of infants born to immigrant mothers living in Ontario, Canada. This work demonstrates reasonable performance of our previously published model to determine gestational age in infants born to immigrants of diverse countries of origin. Our

**Table 2** Cohort characteristics

| | Non-immigrants | India | China | Pakistan | Philippines | Vietnam | Bangladesh | Somalia | Nigeria |
|---|---|---|---|---|---|---|---|---|---|
| Sample size | 230034 | 10038 | 7468 | 5824 | 5441 | 1408 | 1182 | 833 | 800 |
| Birth weight, g | 3382.91 ±558.85 | 3185.42 ±518.73 | 3315.79 ±469.78 | 3208.88 ±504.10 | 3177.88 ±528.21 | 3221.91 ±473.36 | 3114.52 ±508.14 | 3362.01 ±559.40 | 3357.89 ±588.54 |
| Sex | | | | | | | | | |
| Male, n(%) | 118317 (51.43) | 5267 (52.47) | 3860 (51.69) | 3050 (52.37) | 2827 (51.96) | 722 (51.28) | 598 (50.59) | 436 (52.34) | 411 (51.38) |
| Female, n(%) | 111717 (48.57) | 4771 (47.53) | 3608 (48.31) | 2774 (47.63) | 2614 (48.04) | 686 (48.72) | 584 (49.41) | 397 (47.66) | 389 (48.63) |
| Gestational age | 38.84±1.78 | 38.72±1.70 | 38.87±1.51 | 38.73±1.61 | 38.44±1.76 | 38.69±1.58 | 38.54±1.83 | 39.10±1.82 | 38.57±1.89 |
| ≥37 weeks, n(%) | 212989 (92.60) | 9357 (93.22) | 7106 (95.15) | 5447 (93.53) | 4990 (91.71) | 1322 (93.89) | 1097 (92.81) | 775 (93.04) | 744 (93) |
| 32–<37 weeks, n(%) | 15320 (6.66) | 610 (6.08) | 328 (4.39) | 343 (5.89) | 405 (7.44) | 79 (5.61) | 71 (6.01) | 53 (6.36) | 47 (5.88) |
| 28–<32 weeks, n(%) | 1312 (0.57) | 50 (0.5) | 29 (0.39) | 26 (0.45) | 35 (0.64) | 5 (0.36) | 9 (0.76) | 3 (0.36) | 4 (0.5) |
| <28 weeks, n(%) | 413 (0.18) | 21 (0.21) | 5 (0.07) | 8 (0.14) | 11 (0.20) | 2 (0.14) | 5 (0.42) | 2 (0.24) | 5 (0.63) |
| Multigestation pregnancy, n(%) | 7819 (3.40) | 320 (3.19) | 171 (2.29) | 140 (2.40) | 97 (1.78) | 20 (1.42) | 19 (1.61) | 25 (3.00) | 34 (4.25) |
| Low birth weight (<2500g), n(%) | 12716 (5.52) | 787 (7.84) | 307 (4.12) | 410 (7.04) | 427 (7.84) | 72 (5.12) | 100 (8.47) | 48 (5.76) | 51 (6.38) |

findings provide proof-of-principle that metabolic modelling strategies couldbe robust in a variety of international infant cohorts.

The strength of our approach lies in our ability to use population-based datasets to aggregate health and administrative data with ample sample size to evaluate model performance metrics across infant subgroups. In lieu of self-reported race or ethnicity, which were unavailable for our analyses, use of the Canadian Landed Immigrant Database provided data on infants born to immigrant mothers from a diverse range of countries. We acknowledge that naturalised immigrants living in North America may not be representative of individuals living in the country of origin, however, either due to admixture (ie, inter-racial families) or environmental factors (eg, climate, socioeconomic status, diet, sanitation). Indeed, whether subtle variation in model performance in specific immigrant subgroups, such as those from Bangladesh and Somalia, indicate a degree of biological or environmental influence warrants further consideration.

As we work towards evaluating our model in other infant populations, we must consider existing limitations. First, our original approach included a relatively small sample size of preterm infants for model development, which is reflected in decreased accuracy of the model among the most severely preterm infants.[4] We are now working to refine our model to optimise performance across all gestational age categories. Overall precision of metabolic dating methods to within 1–2 weeks compares favourably to other commonly used postnatal dating methods, the accuracy of which vary widely (3–4 weeks)[3 12 13] depending on the method, level of training of the specialist performing the measurements and if the child is small for gestational age. Although limited sample size prevented evaluation of the model among small-for gestational age or low birthweight infants, recent work from our group has demonstrated comparable accuracy among this infant subpopulation.[14] This is particularly important when considering the potential to implement metabolic dating tools in lowand middle income countries given the prevalence of low birthweight infants in low-resource communities. In addition to continuous estimates, provision of gestational age estimates across dichotomous thresholds (eg, 37 weeks gestational age) may be useful for the purpose of population surveillance. Second, currently published models are complex and require data on a large number of analytes measured by mass spectrometry. Ultimately, simplification of models to reduce the number of metabolic variables while maintaining model performance will be required to streamline the approach for scalable, cost-effective applications. An ideal model would include analytes that may be reliably measured from samples obtained immediately after birth (ie, cord blood), and those that are stable through weeks to months of appropriate storage prior to analysis.[14]

**Table 3** Model performance comparison for original validation study and new validation in immigrant and non-immigrant subgroups

| Cohort | RMSE (wks) | Overall | | ≥37 wks | | | 32–<37 wks | | | 28–<32 wks | | | <28 wks | | |
|---|---|---|---|---|---|---|---|---|---|---|---|---|---|---|---|
| | | ±1 wks (%) | ±2 wks (%) | ±1 wks (%) | ±2 wks (%) | | ±1 wks (%) | ±2 wks (%) | | ±1 wks (%) | ±2 wks (%) | | ±1 wks (%) | ±2 wks (%) | |
| Original validation | 1.06 | 66.8 | 94.9 | 69.1 | 96.4 | | 39.0 | 75.6 | | 46.5 | 76.9 | | 50.7 | 77.5 | |
| *Validation based on immigration status in IRCC* | | | | | | | | | | | | | | | |
| New validation cohort (overall) | 1.04 | 67.1 | 95.0 | 69.2 | 96.4 | | 40.1 | 76.3 | | 48.6 | 78.5 | | 48.2 | 79.0 | |
| Non-immigrants | 1.05 | 67.0 | 94.9 | 69.1 | 96.4 | | 39.8 | 76.2 | | 48.4 | 78.43 | | 49.39 | 79.18 | |
| All immigrants | 1.04 | 67.7 | 95.2 | 69.6 | 96.4 | | 41.4 | 76.9 | | 49.5 | 78.9 | | 44.7 | 82.3 | |
| India | 1.04 | 68.1 | 95.2 | 69.8 | 96.3 | | 44.4 | 80.0 | | 42.0 | 70.0 | | 42.9 | 85.7 | |
| China | 0.98 | 71 | 96.2 | 72.9 | 97.6 | | 31.4 | 67.4 | | 51.7 | 75.9 | | 20.0 | 80.0 | |
| Pakistan | 1.05 | 67.2 | 95.3 | 68.4 | 96.2 | | 48.7 | 82.2 | | 61.5 | 88.5 | | 62.5 | 87.5 | |
| Philippines | 1.05 | 66.5 | 94.7 | 69.1 | 96.6 | | 34.8 | 70.9 | | 57.1 | 91.4 | | 45.4 | 100.0 | |
| Vietnam | 1.00 | 70.0 | 95.2 | 72.4 | 96.7 | | 34.2 | 70.9 | | 40.0 | 100.0 | | 0.0 | 100.0 | |
| Bangladesh | 1.14 | 67.4 | 94.6 | 69.1 | 95.9 | | 45.1 | 77.5 | | 44.4 | 77.8 | | 60.0 | 80.0 | |
| Somalia | 1.15 | 62.1 | 93.6 | 64.6 | 95.7 | | 26.4 | 64.2 | | 33.3 | 66.7 | | 50.0 | 100.0 | |
| Nigeria | 1.08 | 65.1 | 94.6 | 66.7 | 95.7 | | 75.0 | 80.8 | | 75.0 | 75.0 | | 40.0 | 80.0 | |

GA, gestational age; IRCC, Citizenship and Immigration Canada's Permanent Resident Database; RMSE, root mean squared error; wks, weeks.

**Table 4** Comparison of model performance for predicting preterm status

| Cohort | <37 weeks GA | | | ≤34 weeks GA | | |
|---|---|---|---|---|---|---|
| | AUC (95% CI) | PPV 80% | p Value* | AUC (95% CI) | PPV 80% | p Value* |
| Original validation | 0.970 (0.966 to 0.974) | – | Reference | 0.991 (0.987 to 0.995) | 80.9 | Reference |
| *Validation based on immigration status in IRCC* | | | | | | |
| New validation cohort (overall) | 0.957 (0.956 to 0.959) | 56.1 | <0.0001 | 0.981 (0.979 to 0.983) | 75.9 | <0.0001 |
| Non-immigrants | 0.958 (0.956 to 0.961) | 57.4 | <0.0001 | 0.981 (0.979 to 0.983) | 75.7 | 0.021 |
| All immigrants | 0.954 (0.95 to 0.958) | 51.7 | <0.0001 | 0.983 (0.978 to 0.987) | 74.3 | 0.0063 |
| India | 0.947 (0.936 to 0.959) | 49.3 | 0.00027 | 0.980 (0.968 to 0.993) | 61.8 | 0.240 |
| China | 0.957 (0.943 to 0.972) | 42.0 | 0.092 | 0.983 (0.966 to 1.00) | 86.3 | 0.78 |
| Pakistan | 0.949 (0.933 to 0.964) | 43.1 | 0.010 | 0.960 (0.936 to 0.985) | 77.0 | 0.103 |
| Philippines | 0.958 (0.945 to 0.971) | 57.9 | 0.082 | 0.992 (0.982 to 1.00) | 69.4 | 0.505 |
| Vietnam | 0.949 (0.916 to 0.981) | 48.9 | 0.209 | 0.991 (0.965 to 1.02) | 88.2 | 0.840 |
| Bangladesh | 0.945 (0.911 to 0.979) | 49.3 | 0.151 | 0.947 (0.888 to 1.00) | 69.6 | 0.300 |
| Somalia | 0.927 (0.880 to 0.974) | 30.9 | 0.072 | 0.974 (0.924 to 1.02) | 83.3 | 1.000 |
| Nigeria | 0.964 (0.930 to 0.998) | 51.7 | 0.733 | 0.965 (0.910 to 1.02) | 100.0 | 0.369 |

*p Value for two-sided z-test for the difference between original and subgroup AUC.
AUC, area under the receiver operating characteristic curve; GA, gestational age; IRCC, Citizenship and Immigration Canada's Permanent Resident Database; PPV, positive predictive value.

In summary, our results provide reassurance that an algorithm developed from an Ontario-based population performs consistently well across a range of ethnicities. Ultimately, further validation studies will be required to evaluate the performance of this and other postnatal dating models in infants born and living across a range of international settings to determine if a single global algorithm or multiple regional algorithms should be derived.

**Contributors** SH and RD were involved in data acquisition and statistical analysis. SH, MSQM and KW drafted and edited the manuscript. KMA and MSQM provided project coordination. BKP and PC critically edited the manuscript for important intellectual content. KW was responsible for the conceptual design of the study.

**Funding** This work was supported by The Bill & Melinda Gates Foundation [OPP1141535]. It was also supported by the Institute for Clinical Evaluative Sciences (ICES), which is funded by an annual grant from the Ontario Ministry of Health and Long-Term Care (MOHLTC).

**Competing interests** All authors have completed the ICMJE uniform disclosure form at www.icmje.org/coi_disclosure.pdf and declare: the authors had financial support from The Bill & Melinda Gates Foundation for the submitted work; no financial relationships with any organisations that might have an interest in the submitted work in the previous three years; no other relationships or activities that could appear to have influenced the submitted work. The opinions, results and conclusions reported in this paper are those of the authors and are independent from the funding sources. No endorsement by ICES or the Ontario MOHLTC is intended or should be inferred. Parts of this material are based on data and information compiled and provided by the Canadian Institute of Health Information (CIHI). However, the analyses, conclusions, opinions and statements expressed herein are those of the author, and not necessarily those of CIHI.

**Ethics approval** Ottawa Health Science Network Research Ethics Board.

**Provenance and peer review** Not commissioned; externally peer reviewed.

**Data sharing statement** All data used in this study were obtained from the Institute for Clinical Evaluative Sciences and are accessible to individuals with appropriate authorisation.

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
