## [Reviewer comments · BMJ Open]

ARTICLE DETAILS

TITLE (PROVISIONAL)	Performance of a postnatal metabolic gestational age algorithm: a retrospective validation study among ethnic subgroups in Canada
AUTHORS	Hawken, Steven; Ducharme, Robin; Murphy, Malia; Atkinson, Katherine; Potter, Beth; Chakraborty, Pranesh; Wilson, Kumanan

VERSION 1 – REVIEW

REVIEWER	Prakesh Shah University of Toronto, Canada
REVIEW RETURNED	10-Jan-2017

GENERAL COMMENTS	1. I am unconvinced about the value of paper that describes development of a model that predicts GA in which for real life application of the results that authors have presented will require a long wait, ability to conduct these assays, incorporate results in a model and the. Receive answer which will be within 2 week range of accuracy. How does this help anyone? Clinicians? Policy makers? The real utility of any such prediction will lie only if the method allows me to diagnose a at risk neonate when he or she is born. I am not sure how and what will be clinical implications of this project. 2. Secondly, whether this model is good or not can only be confirmed if this is cross validated with other method of estimating GA. For those with unknown GA we use modified Ballard to estimate GA right at birth and has 1 week of error margin. Why do one need to go through such painstaking exercise when clinical exam can be completed in a matter of 10-15 minutes. I am really struggling with clinical implications of these findings. Minor: the description about Gates foundation aims in introduction and discussion is out of place for a manuscript.
--

REVIEWER	Lizelle Van Wyk Neonatologist University of Stellenbosch Cape Town South Africa
REVIEW RETURNED	23-Jan-2017

GENERAL COMMENTS	The following needs clarification please Introduction: The authors have not included, mentioned or referenced the accuracy and availability of the various current gestational age methodologies in current use- ultrasound, LMP, fundal height, clinical parameters (Ballard/ Dubowitz scores) foot length, etc. As the authors aim is to offer NBS (newborn blood spot) as a viable alternative to existing methods in low resource settings this is of importance. The implications of using the NBS in low resource settings also needs to be discussed or mentioned. One of the implications of NBS is the cost for a population with a high birth rate (as in most low resource countries) and the availability of facilities to follow-up and treat the diagnosed abnormality. Although this is not the aim of the research, this factor needs to be taken into consideration. Methodology: What levels of the analytes were used - cut-off values should be included (or referenced) to allow international readers to evaluate how levels may differ between countries. TSH can be used as an example as different laboratories in the same country have been shown to use different levels. Results Some idea needs to be given of the population demographics. If it is the same as reference 3 then there is a very small percentage of preterm, very low birth weight infants and SGA infant in the cohort. This may impact on the applicability of the study in a low resource setting. Although the grouping for prematurity (<37 and <34 weeks) is understandable from a statistical point of view it is not of clear clinical reference. This should be discussed by the authors. Why was a group <32 or even <28 weeks not evaluated as this would be of clinical importance Statistics The appropriateness of AUC for a multitude of tests is not clearly understood. This may require some statistical input. The +/- 1 and 2 week brackets should preferably be performed for the whole group (as shown) as well for the <37 and <34 weeks as the model may lose accuracy as the gestational age decreases. This would be of clinical importance Discussion If the authors aim to present this as a low resource test then some discussion is required regarding availability (short and long term) of these test in the various countries, as well as its cost effectiveness (eg a LMP or foot length still remains much cheaper than an NBS) and the long-term implications of NBS testing In essence, this submission remains a "validity" question, which I think the authors have adequately addressed. The references to using it in low resource should, for now, be removed, as it clouds the aim of the study - proving the validity of the model in different ethnic subgroups. Pending further research, as stated by the authors
---

REVIEWER	Anne-Laure Boulesteix (professor) LMU Munich, Germany
REVIEW RETURNED	10-Mar-2017

GENERAL COMMENTS	Note: My report is focused on statistical aspects. I am not able to
---

fully assess other aspects.

As far as I can judge, the statistical analysis is correct. Here are some suggestions to improve the paper (some of them important, some of them minor), which should be relatively easy to take into account by the authors.

1. From the abstract, it is not clear which prediction algorithm the authors aim to validate. Even if it becomes clear later when reading the paper (and even if it is implicitly suggested by the term "validation study"), I think it would be good to state in the abstract that the investigated prediction algorithm has been developed based on an earlier cohort data and published elsewhere.

2. p2 line 46: What is the unit of the RMSE?

3. Until p3 line 3 it is not clear (at least to me as a statistician) that the study is concerned with *postnatal* determination of gestational age rather than with pre-natal prediction for the purpose of prematurity prevention. Please clarify this point earlier in the abstract, for example by simply adding the term "postnatal".

4. p7 line 5-6: What does "a total of 311 model terms" mean? Please clarify.

5. I think it would be good to include the model coefficients (for linear and logistic regression) somewhere in the paper, either in the main text or as additional file.

6. Which statistical tool (SAS, STATA, R,...?) for data analysis?

7. p8 line 32-34: "Original model performance characteristics are provided for comparison": here the authors mean the performance on the previous validation cohort as presented in Ref (3), but I think it would be worth saying it explicitly.

8. p8: "Absolute prediction was accurate to within ± 1 week in continuous linear regression models". This sentence is not very clear to me. Does it refer to the fact that the RMSE is approximately 1? One could also understand it as "all predictions are within the interval "real value ± 1 week" (although it would be obviously quite unrealistic and contradictory with the table).

9. p8: The term RMSE (root mean squared error) is clear to me, but is it clear to all potential readers of the paper?

10. p8: The unit should be added whenever an RMSE is mentioned.

11. p11: "linear regression analysis": please replace by the more common phrase "linear regression analysis".

12. Which method was used to derive confidence intervals for the AUC?

13. I think it would be very helpful to also report prediction accuracy for the whole validation cohort (i.e. immigrants and non-immigrants

	together) for direct comparison with the original cohort, i.e. to see if application of the model to later data leads to performance deterioration. 14. p4 line 25: typo "of of". 15. p4 line 46: aforementioned 16. p8 line 37-41: synthax problem?
--	--

REVIEWER	Ulla Sovio University of Cambridge, United Kingdom
REVIEW RETURNED	17-Mar-2017

GENERAL COMMENTS	The authors have sought to validate a previously developed gestational age prediction algorithm in ethnic subgroups. I have done a statistical review of the manuscript by assessing it against the guidelines published by Altman et al (BMJ 2009;338:b605), Collins et al (BMJ 2015;350:g7594) and Riley et al (BMJ 2016;353:i3140). The authors employed the multivariable linear regression model coefficients derived from their original study (Reference 3) and then compared the gestational ages (GAs) predicted by the model with actual GAs, which is correct. In parts, the reporting of the method and the results is incomplete and I have a few suggestions on how to improve the manuscript. Study Design: The authors should add how the actual GA was determined (this is mentioned in Reference 3 but there is no mention of it in the manuscript). The % of babies whose GA was based on solely on the last menstrual period would be of interest, and if the availability of ultrasound measurement differed by immigration status. Model Validation: The authors could describe more clearly how logistic regression was applied in the context of validation. I assume that the GA calculated using the original model coefficients was then used as an exposure in the model that predicts actual GA categorised by prematurity. The authors could also say explicitly which model performance characteristics they calculated and how they dealt with missing data. Results: The authors could present a descriptive table of the validation cohort by immigration status, which would allow the reader to compare the distributions in the validation cohort with the distributions in the original cohort where the model was developed (Reference 3 Table 2). The authors could also describe the proportion of missing data. Table 2: The number of events in some of the immigrant categories
---

	is rather low (see Riley et al p. 2). I would suggest that the authors also present the results in a combined group of all immigrants. Addition of model calibration plots and statistics would improve the paper. Discussion: In the current manuscript, the authors did not update the model coefficients or recalibrate the model to ethnic subpopulations. This could be discussed in more detail, as there is currently only a brief mention about further refining the prediction algorithms by development of ethnicity-specific models
--	--

VERSION 1 – AUTHOR RESPONSE

Reviewer 1

Reviewer Name: Prakesh Shah

Institution and Country: University of Toronto, Canada

Competing Interests: None declared

1. I am unconvinced about the value of paper that describes development of a model that predicts GA in which for real life application of the results that authors have presented will require a long wait, ability to conduct these assays, incorporate results in a model and the. Receive answer which will be within 2-week range of accuracy. How does this help anyone? Clinicians? Policy makers? The real utility of any such prediction will lie only if the method allows me to diagnose a at risk neonate when he or she is born. I am not sure how and what will be clinical implications of this project.

The reviewer is correct that this is an interim study to further validate the algorithm before testing it in a live setting. Based on the comments from the reviewers, we have removed most references to using metabolic dating algorithms in low resource settings, as it clouds the aim of our study - proving the validity of the model in different ethnic subgroups.

We now highlight this as a validation, proof-of-concept exercise throughout the introduction and the discussion of the manuscript and address existing limitations to the approach that will need to be addressed before clinical application.

Discussion: “As we work towards evaluating our model in other infant populations, we must consider existing limitations to the published model. First, our original approach included a relatively small sample size of preterm infants for model development, which resulted in decreased accuracy amongst the most severely preterm infants. We are now working to refine our model to optimize performance amongst all gestational age categories. Second, currently published models are complex, and require data on a large number of analytes measured by mass spectrometry. Ultimately, simplification of models to reduce the number of metabolic variables while maintaining model performance will be required to streamline the approach for potential application in the field.”

2. Secondly, whether this model is good or not can only be confirmed if this is cross validated with other method of estimating GA. For those with unknown GA we use modified Ballard to estimate GA right at birth and has 1 week of error margin. Why do one need to go through such painstaking exercise when clinical exam can be completed in a matter of 10-15 minutes. I am really struggling

with clinical implications of these findings.

Thank you for this comment. Postnatal scoring systems based on physical and neurological criteria such as the Ballard and Dubowitz methods are indeed commonly employed in LMIC where antenatal estimates are unavailable. Validation of these approaches have highlighted their limitations, however. Postnatal scoring systems require advanced clinical training, and even then, still subject to variability based on the individual applying the test. Recent work has determined that these approaches are accurate to no more than within 3-4 weeks of ultrasound validated gestational age, suggesting that they can actually perform more poorly than estimates based on birthweight alone. The introduction has been re-framed to highlight the limitations of currently available postnatal dating methods.

Introduction: “Alternative antenatal dating methods including last menstrual period and fundal height measurements are hampered by poor recall history, and prevalence of growth restriction, whereas commonly used postnatal assessments that score infants on neurologic and physical criteria only accurate to within 3-4 weeks of true, ultrasound-validated gestational age.”

As above, based on the comments from the reviewers, we have removed most references to using metabolic dating algorithms in low resource settings, as it clouds the aim of our study - proving the validity of the model in different ethnic subgroups. We now highlight this as a validation, proof-of-concept exercise throughout the introduction and the discussion of the manuscript and address existing limitations to the approach that will need to be addressed before clinical application.

Discussion: “As we work towards evaluating our model in other infant populations, we must consider existing limitations to the published model. First, our original approach included a relatively small sample size of preterm infants for model development, which resulted in decreased accuracy amongst the most severely preterm infants. We are now working to refine our model to optimize performance amongst all gestational age categories. Second, currently published models are complex, and require data on a large number of analytes measured by mass spectrometry. Ultimately, simplification of models to reduce the number of metabolic variables while maintaining model performance will be required to streamline the approach for potential application in the field.”

3. Minor: the description about Gates foundation aims in introduction and discussion is out of place for a manuscript.

Thank you, this description has been removed from the manuscript.

Reviewer 2

Reviewer Name: Lizelle Van Wyk

Institution and Country: Neonatologist, University of Stellenbosch, Cape Town, South Africa

Competing Interests: None

1. The authors have not included, mentioned or referenced the accuracy and availability of the various current gestational age methodologies in current use- ultrasound, LMP, fundal height, clinical parameters (Ballard/ Dubowitz scores) foot length, etc. As the authors aim is to offer BNS (newborn blood spot) as a viable alternative to existing methods in low resource settings this is of importance.

Thank you, we acknowledge the use of dating methods in low-resource areas, and have included mention of antenatal and postnatal alternatives for gestational dating in the introduction of the manuscript.

Introduction: “Alternative antenatal dating methods including last menstrual period and fundal height measurements are hampered by poor recall history, and prevalence of growth restriction, whereas commonly used postnatal assessments that score infants on neurologic and physical criteria only

accurate to within 3-4 weeks of true, ultrasound-validated gestational age.”

2. The implications of using the NBS in low resource settings also needs to be discussed or mentioned. One of the implications of NBS is the cost for a population with a high birth rate (as in most low resource countries) and the availability of facilities to follow-up and treat the diagnosed abnormality. Although this is not the aim of the research, this factor needs to be taken into consideration.

Thank you for this comment. At the request of this Reviewer (see subsequent comments), we have removed content related to applications in low resource settings, as it may be distracting to readers. Rather, we have more appropriately reframed the manuscript as a validation exercise. We have however, highlighted within the discussion a need to address existing limitations, including simplification of the models for potential application in low resource settings:

In a previous paper we have discussed the practicality of NBS in low resource settings. This paper has been referenced in the manuscript:

Wilson K, Hawken S, Murphy MS, et al. Postnatal Prediction of Gestational Age Using Newborn Fetal Hemoglobin Levels. *EBioMedicine*. 2017;15:203-9.

Discussion: “As we work towards evaluating our model in other infant populations, we must consider existing limitations to the published model. First, our original approach included a relatively small sample size of preterm infants for model development, which resulted in decreased accuracy amongst the most severely preterm infants. We are now working to refine our model to optimize performance amongst all gestational age categories. Second, currently published models are complex, and require data on a large number of analytes measured by mass spectrometry. Ultimately, simplification of models to reduce the number metabolic variables while maintaining model performance will be required to streamline the approach for potential application in the field. An ideal model would be useful with analytes that are valid immediately after birth, can be measured using simple assays, and can be determined from cord blood.

3. What levels of the analytes were used - cut-off values should be included (or referenced) to allow international readers to evaluate how levels may differ between countries. TSH can be used as an example as different laboratories in the same country have been shown to use different levels.

We did not use any clinical/screening cutoffs in our analysis, with the exception of excluding infants that screened positive for any newborn screening condition (ie. exhibited analyte levels orders of magnitude different from the distribution of screen negative infants). The reason for exclusion of these infants was that their metabolic profiles would have significantly distorted our statistical modelling approaches, had they been included. All analyte values underwent a normalizing/variance stabilizing log transformation, followed by standardization in order that the beta coefficient for each analyte in the model reflected the effect of a 1 SD change in that analyte.

4. Some idea needs to be given of the population demographics. If it is the same as reference 3 then there is a very small percentage of preterm, very low birth weight infants and SGA infants in the cohort. This may impact on the applicability of the study in a low resource setting.

Thank you, we have now included a table (Table 2) of demographics in the manuscript with rates of preterm birth, sex, multiple gestations, as well mean and SD of birthweight and gestational age.

5. Although the grouping for prematurity (<37 and <34 weeks) is understandable from a statistical

point of view it is not of clear clinical reference. This should be discussed by the authors. Why was a group <32 or even <28 weeks not evaluated as this would be of clinical importance

Thank you for this observation.

We have included a new table that summarizes the demographics of our cohort, including distribution across clinical gestational age categories (Table 2). Our purpose for evaluating the performance of our regression model in infant subgroups (<37 and ≤34 weeks) are provided within the text of the manuscript:

“Preterm birth thresholds were <37 weeks gestational age, the clinical threshold for preterm birth, and ≤34 weeks gestational age, a threshold that represents the lower limit of the late preterm period. Infants born ≤34 weeks gestational age (early preterm and severe preterm infants) are at increased health risk compared to those born late preterm or at term.”

6. The appropriateness of AUC for a multitude of tests is not clearly understood. This may require some statistical input.

We thank the reviewer for their careful consideration of our approach. To clarify, the AUC is useful in assessing the discrimination of a logistic regression model, however it is not a sensitive way of comparing the performance of competing models or across populations. We have therefore tried to present a number of different metrics of model performance, as no single metric tells the entire story. Therefore we have presented RMSE, which gives average deviation in the original units-gestational age in weeks, as well as absolute agreement within 1 week and within 2 weeks gestational age. We also presented the AUC from logistic regression models classifying infants in clinically important groups. The AUC is also not enough to fully describe the model performance, because it is a summary across all possible cutoffs. We therefore also present the PPV for fixed sensitivity of 80% (which gives a clear sense of how many false positives would be included if the cutoff was set such that 80% of true premature infants were captured by the test cutoff).

7. The +/- 1 and 2 week brackets should preferably be performed for the whole group (as shown) as well for the <37 and <34 weeks as the model may lose accuracy as the gestational age decreases. This would be of clinical importance

We agree. We now provide model performance overall as well as across clinical reference categories for prematurity in Table 3.

8. If the authors aim to present this as a low resource test then some discussion is required regarding availability (short and long term) of these test in the various countries, as well as its cost effectiveness (eg a LMP or foot length still remains much cheaper than an NBS) and the long-term implications of NBS testing)

As per the following comment from this same reviewer, we have opted to minimize this content in the manuscript. We have however added mention of currently used methods for gestational age dating, including LMP dating, fundal height and postnatal scoring methods.

Introduction: “Knowledge of gestational age at the time of birth is vital for ensuring adequate provision of newborn care, assessing neurocognitive function and tracking of developmental milestones.(1, 2) For some pregnancies, where antenatal ultrasound screening is not available, such as those occurring in jurisdictions with challenging socioeconomic conditions and/or limited access due to rurality, determination of gestational age can be challenging. Alternative antenatal dating methods including last menstrual period and fundal height measurements are hampered by poor recall history, and prevalence of growth restriction, whereas commonly used postnatal assessments that score

infants on neurologic and physical criteria only accurate to within 3-4 weeks of true, ultrasound-validated gestational age. World health and philanthropic organizations are now seeking novel ways of determining gestational age, both to improve individual care, and to provide reliable, population-based estimates of preterm birth.”

As identified in by the reviewer in the following comment, the main objective of this study was the validation of our algorithm, rather than exploring the scalability/cost-effectiveness of our approach. We have included indication that future work should include simplification of metabolic dating models for potential application in clinical settings:

Discussion: “Second, currently published models are complex, and require data on a large number of analytes measured by mass spectrometry. Ultimately, simplification of models to reduce the number metabolic variables while maintaining model performance will be required to streamline the approach for potential application in the field.”

9. In essence, this submission remains a "validity" question, which I think the authors have adequately addressed. The references to using it in low resource should, for now, be removed, as it clouds the aim of the study - proving the validity of the model in different ethnic subgroups. Pending further research, as stated by the authors

We agree with the reviewer, that the manuscript needs to be reframed to minimize justifications of application in low resource settings:

Discussion: “This study validates a postnatal gestational age estimation model in subgroups of infants born to immigrant mothers in Ontario, Canada. Examination of performance characteristics demonstrated reasonable performance of our previously published model to determine gestational age in infants born to immigrants of diverse ethnicities/countries of origin. These findings provide proof-of-principle that metabolic modeling strategies will be robust in a variety of international infant cohorts.....

... Our results provide reassurance that an algorithm developed from an Ontario-based population performs consistently well across a range of ethnicities. As algorithms for postnatal determination of gestational age may be further refined by development of ethnicity-specific models, we are currently undertaking a series of validation studies to evaluate the performance of our model in infants born and living across a range of international settings.”

Reviewer 3

Reviewer Name: Anne-Laure Boulesteix (professor)

Institution and Country: LMU Munich, Germany

Competing Interests: None declared

1. From the abstract, it is not clear which prediction algorithm the authors aim to validate. Even if it becomes clear later when reading the paper (and even if it is implicitly suggested by the term "validation study"), I think it would be good to state in the abstract that the investigated prediction algorithm has been developed based on an earlier cohort data and published elsewhere.

Thank you. The abstract text has been revised for clarification:

Abstract, Objectives: “Validation of published models have previously been limited to cohorts consisting of infants of largely white-caucasian ethnicity. In this study, we sought to determine the validity of a published gestational age estimation algorithm among recent immigrants to Canada where maternal landed immigrant status was used as a surrogate measure of infant ethnicity.”

2. p2 line 46: What is the unit of the RMSE?

The unit of RMSE is 'weeks'. This is now clarified in the text.

3. Until p3 line 3 it is not clear (at least to me as a statistician) that the study is concerned with *postnatal* determination of gestational age rather than with pre-natal prediction for the purpose of prematurity prevention. Please clarify this point earlier in the abstract, for example by simply adding the term "postnatal".

Thank you, we have clarified this earlier within the 'Objectives' of the abstract:

Abstract, Objectives: "Biological modelling of routinely collected newborn screening data has emerged as a novel method for deriving postnatal gestational age estimates. Validation of published models have previously been limited ..."

4. p7 line 5-6: What does "a total of 311 model terms" mean? Please clarify.

Thank you, this means that the regression model included a total of 311 parameter estimates, including main effects, quadratic and cubic effects plus interaction terms. We clarified this within the text:

Materials and Methods

Original model: "The final regression model included a total of 311 parameter estimates, including main effects, quadratic and cubic effects plus interaction terms."

5. I think it would be good to include the model coefficients (for linear and logistic regression) somewhere in the paper, either in the main text or as additional file.

Thank you. The full set of model coefficients was published in our original report of the model (K Wilson et al. AJOG 2016), and has been referenced throughout the manuscript. If the reviewer and editors feel strongly the model terms should be included in this paper as well, either in the methods section or in supplementary materials, we can provide this.

6. Which statistical tool (SAS, STATA, R,...?) for data analysis?

The statistical tool used SAS 9.4 (Cary, NC). This has been included within the methods section.

7. p8 line 32-34: "Original model performance characteristics are provided for comparison": here the authors mean the performance on the previous validation cohort as presented in Ref (3), but I think it would be worth saying it explicitly.

Thank you. We have clarified this as "from the previous validation cohort".

8. p8: "Absolute prediction was accurate to within ± 1 week in continuous linear regression models". This sentence is not very clear to me. Does it refer to the fact that the RMSE is approximately 1? One could also understand it as "all predictions are within the interval "real value ± 1 week" (although it would be obviously quite unrealistic and contradictory with the table).

Thank you we have clarified this in the text:

"Absolute gestational age estimation was on average accurate to within 1 week in continuous linear

regression models (RMSE~1 week)".

9. p8: The term RMSE (root mean squared error) is clear to me, but is it clear to all potential readers of the paper?

Thank you. We have added a sentence to clarify this detail :

Methods: "The deviation of each calculated gestational age from the true gestational age of each infant is the residual model error for that infant (in unit, weeks). The residual model error can be positive or negative depending on the direction of the difference. The MSE is the sum of each of those residual errors after squaring it (also rendering all values positive) for all infants. MSE is in the units of (weeks)². Taking the square root of the MSE yielded an overall "average deviation" in unit, weeks. An RMSE of 1 may be interpreted as the regression model estimating gestational age within ± 1 week on average."

10. p8: The unit should be added whenever an RMSE is mentioned.

Thank you. We have included the unit of RMSE (weeks) when mentioned in the text.

11. p11: "linear regression analysis": please replace by the more common phrase "linear regression analysis".

Thank you, we have made this change, accordingly.

12. Which method was used to derive confidence intervals for the AUC?

We used the methods described by Hanley and McNeil to derive the confidence intervals for AUC. We have added this information to the materials and methods section, and references.

Hanley and McNeil. The Meaning Use of the Area under the receiver operator characteristic(ROC) curve. Radiology. 1982 Apr;143(1):29-36.

13. I think it would be very helpful to also report prediction accuracy for the whole validation cohort (i.e. immigrants and non-immigrants together) for direct comparison with the original cohort, i.e. to see if application of the model to later data leads to performance deterioration.

Thank you. In response to this comment and those of another reviewer, we have now added details for an analysis of overall immigrants, and overall immigrants plus non-immigrants.

14. p4 line 25: typo "of of".

We have corrected this error.

15. p4 line 46: aforementioned

Thank you, this have been corrected.

16. p8 line 37-41: synthax problem?

Thank you, this have been corrected.

Reviewer 4

Reviewer Name: Ulla Sovio

Institution and Country: University of Cambridge, United Kingdom

Competing Interests: None declared

The authors have sought to validate a previously developed gestational age prediction algorithm in ethnic subgroups. I have done a statistical review of the manuscript by assessing it against the guidelines published by Altman et al (BMJ 2009;338:b605), Collins et al (BMJ 2015;350:g7594) and Riley et al (BMJ 2016;353:i3140).

The authors employed the multivariable linear regression model coefficients derived from their original study (Reference 3) and then compared the gestational ages (GAs) predicted by the model with actual GAs, which is correct. In parts, the reporting of the method and the results is incomplete and I have a few suggestions on how to improve the manuscript.

Study Design

1. The authors should add how the actual GA was determined (this is mentioned in Reference 3 but there is no mention of it in the manuscript). The % of babies whose GA was based on solely on the last menstrual period would be of interest, and if the availability of ultrasound measurement differed by immigration status.

The GA was based on best obstetrical estimate although data were unavailable to discriminate those estimates that were based on dating ultrasounds versus last menstrual period. This has been clarified within the manuscript:

“Gestational age was based on best obstetrical estimate (last menstrual period, dating ultrasound, or a combination)”

Model Validation

2. The authors could describe more clearly how logistic regression was applied in the context of validation. I assume that the GA calculated using the original model coefficients was then used as an exposure in the model that predicts actual GA categorised by prematurity. The authors could also say explicitly which model performance characteristics they calculated and how they dealt with missing data.

Thank you – to ensure that the methods of model validation are clear to the reader, we have added additional detail in the methods as follows:

Methods: “The coefficients from the reference linear regression model were fixed, and used to score the validation cohort. Calculated gestational age was used as the independent variable in logistic regression models to estimate the dichotomous categories of prematurity, for which AUC and sensitivity, specificity and positive predictive value were calculated.”

Results:

3. The authors could present a descriptive table of the validation cohort by immigration status, which would allow the reader to compare the distributions in the validation cohort with the distributions in the original cohort where the model was developed (Reference 3 Table 2). The authors could also describe the proportion of missing data.

Thank you for this suggestion. We have included a table of descriptives for infants overall and for country subgroups in response to this comment and that of another reviewer. We have also included a description of the proportion of missing data:

Methods: “Missing data occurred for a very proportion of the individual analytes used in the prediction

model (<1%). Because all a full panel of analyte data per infant was required to score new observations, only infants with complete records for all analytes and other covariates were included in this validation exercise. The infants excluded due to missing gestational age, analytes or other covariates constituted <5% of the cohort.”

4. Table 2: The number of events in some of the immigrant categories is rather low (see Riley et al p. 2). I would suggest that the authors also present the results in a combined group of all immigrants.

Although pooling immigrants in smaller subgroups would improve the sample size, it would result in adding together potentially heterogeneous groups of infants, which would make interpretation problematic. Because we are scoring infants using an existing model, and not building a new regression model, the issues around sample size versus stability of parameter estimates are not a significant concern in this setting.

5. Addition of model calibration plots and statistics would improve the paper.

Thank you for this suggestion. While calibration plots and statistics would lend to the statistical depth of the paper, the submitted manuscript was intended as a short report, and calibration plots/ slopes/ intercepts would require a great deal more space to present in this manuscript, and likely distract from the aims of the study. If the reviewers and editors feel strongly these should be included we will do so, and anticipate that the additional plots/statistics would be included as an online supplement.

Discussion:

6. In the current manuscript, the authors did not update the model coefficients or recalibrate the model to ethnic subpopulations. This could be discussed in more detail, as there is currently only a brief mention about further refining the prediction algorithms by development of ethnicity-specific models.

We have not updated model coefficients or applied calibration adjustments in the validation models fit in the current manuscript, as the population presented here is not a true ‘external population’. Ultimately, when we do validate the model in external infant populations, we do anticipate that model recalibrations and/or coefficients updates will optimize model performance. As in comment 5 from this same reviewer, we can produce calibration plots/statistics if the reviewers/editors feel strongly they need to be here.

VERSION 2 – REVIEW

REVIEWER	Lizelle Van wyk University of Stellenbosch Cape Town South Africa
REVIEW RETURNED	24-May-2017

GENERAL COMMENTS	Thank you to the authors for this revision Please check sentence structures and wording of the changes - especially pg29 The inserted paragraph on pg 29 is also contradictory - missing data is initially stated as <1% and then later in the paragraph as <5% - this needs resolution As stated by the authors in the introduction LMP is a flawed method
---

	of accurately defining gestational age; yet it is used in this research. This does not make sense - it would be much more appropriate to limit GA to ultrasound dating, which is the most accurate dating method currently available. This would ensure an accurate comparison with which to compare the metabolic models The dichotomisation of gestational age, although statistically useful, is not clinically useful. Management is determined by gestational age in many countries and a gestational age <34 weeks is not a clinical helpful estimation of gestational age. The dichotomisation also does not correspond to the data presented in tables 2 & 3 - no 34 week cutoff is included - this makes the data a bit difficult to comprehend. It would be clinically more useful to use the gestational age categories in tables 2/3 rather than those of <34, 34-37, >37, as in table 3 pg 31 "probability of being born preterm" - the regression will be able to predict the probability of prematurity - not "being born premature" A limitation that needs to be discussed is IUGR/ SGA and its possible effect on the model as IUGR/ SGA represents a significant risk factor to GA estimation It is stated in the introduction that alternative methods of GA estimation are being sought. Alternative methods are not discussed anywhere in this research paper - foot lengths, cranial ultrasound, etc. Alternative methods' accuracy are also not compared to the current metabolic models. The cost of metabolic screening is excessive in most non-American/ European countries and often not even available in lower/ middle income countries. This aspect has not been mentioned in the study - it is thus difficult to accept the clinical applicability of this model in these countries
--	--

REVIEWER	Anne-Laure Boulesteix LMU Munich, Germany
REVIEW RETURNED	16-May-2017

GENERAL COMMENTS	I just have two remarks about the (R)MSE:  - "The MSE is the sum of each of those residual errors" -> please replace the word "sum" by "mean". - "An RMSE of 1 may be interpreted as the regression model estimating gestational age within ± 1 week on average": This statement is confusing. I recommend removing it (in the methods section and whenever it occurs in the Results). Typo: "Missing data occurred for a very proportion": a word is missing.
--

VERSION 2 – AUTHOR RESPONSE

1. Please check sentence structures and wording of the changes - especially pg29

Author Response: Thank you. The manuscript has been extensively reviewed, and edited to ensure that issues in grammar, spelling and concept ambiguity have been resolved.

2. The inserted paragraph on pg 29 is also contradictory - missing data is initially stated as <1% and then later in the paragraph as <5% - this needs resolution

Author Response: Thank you. The 1% missing data refers to missing analyte data, where as the missing 5% refers any infant excluded for missing covariates (analytes, gestational age etc). To remove any potential confusion the text has been revised:

Materials and Methods – Original Model: “Because all a full panel of analyte data per infant was required to score new observations, only infants with complete records for all analytes and other covariates were included in this validation exercise. The infants excluded due to missing covariates including gestational age, analytes or others constituted <5% of the cohort.”

3. As stated by the authors in the introduction LMP is a flawed method of accurately defining gestational age; yet it is used in this research. This does not make sense - it would be much more appropriate to limit GA to ultrasound dating, which is the most accurate dating method currently available. This would ensure an accurate comparison with which to compare the metabolic models.

Author Response: Thank you. We agree that limiting our analyses to those pregnancy with ultrasound confirmed gestational ages would have been optimal. Unfortunately the datasets used in this study did not permit us to identify those women who’s best obstetrical estimate was determine via ultrasound dating vs LMP vs combination. Policy briefs from the Institute for Clinical Evaluative Sciences however indicate that >99% of women in the province of Ontario receive at least one ultrasound (dating or otherwise) during the course of their pregnancies. We have now included this information within the manuscript.

Material and Methods – Original Model: “Gestational age was based on best obstetrical estimate (last menstrual period, dating ultrasound, or a combination). Note: >99% of women in Ontario receive at least one ultrasound during the course of pregnancy.”

4. The dichotomization of gestational age, although statistically useful, is not clinically useful. Management is determined by gestational age in many countries and a gestational age <34 weeks is not a clinical helpful estimation of gestational age. The dichotomization also does not correspond to the data presented in tables 2 & 3 - no 34 week cutoff is included - this makes the data a bit difficult to comprehend. It would be clinically more useful to use the gestational age categories in tables 2/3 rather than those of <34, 34-37, >37, as in table 3

Author Response: Thank you for this comment. We also believe that provision of clinically relevant gestational age categories is the most useful to readers. For this reason, Tables 2 (descriptive characteristics) and 3 (model performance; linear regression) both present the standard clinical thresholds used to categorize infants across preterm categories.

Our justification to providing model performance characteristics using dichotomous thresholds of preterm birth are provided in the materials and methods:

“Preterm birth thresholds were <37 weeks gestational age, the clinical threshold for preterm birth, and ≤34 weeks gestational age, a threshold that represents the lower limit of the late preterm period. Infants born ≤34 weeks gestational age (early preterm and severe preterm infants) are at increased health risk compared to those born late preterm or at term.”

As alluded to by the reviewer, estimation of gestational age across dichotomous thresholds may be particularly useful for statistical or administrative purposes (eg. providing population level estimates), and we have expanded on this point within the discussion:

Discussion: "As we work towards evaluating our model in other infant populations, we must consider existing limitations. First, our original approach included a relatively small sample size of preterm infants for model development, which resulted in decreased accuracy amongst the most severely preterm infants⁴. We are now working to refine our model to optimize performance amongst all gestational age categories. Precision of metabolic dating methods to within 1-2 weeks compares favourably to other commonly used postnatal dating methods, the accuracy of which vary widely (3-4 weeks) depending on the method, level of training of the specialist performing the measurements and if the child is small for gestational age....In addition to continuous estimates, provision of gestational age estimates across dichotomous thresholds (eg ≥ 37 or < 37 weeks gestational age) may be useful for the purpose of population surveillance."

We are open to removing the model performance characteristics for determining >34 weeks or ≤ 34 weeks gestational age from Table 4 however, should the reviewer and/or editor feel this data are cumbersome or confusing.

5. pg 31 "probability of being born preterm" - the regression will be able to predict the probability of prematurity - not "being born premature"

Author Response: We thank the reviewer for her careful review of the manuscript. The above noted change has been made: "...probability of preterm birth".

6. A limitation that needs to be discussed is IUGR/ SGA and its possible effect on the model as IUGR/ SGA represents a significant risk factor to GA estimation

Author Response: Thank you, we agree that consideration of how well metabolic dating algorithms perform amongst SGA infants is particularly important as it is amongst SGA and low birthweight infants that current methods of postnatal dating are particularly limited. Indeed, currently used postnatal scoring systems all underperform in this unique population. Although limited sample sizes (overall and of SGA infants) prevented us from evaluating the model in this particular cohort, recent work from our group has demonstrated that various iterations of our model are capable of providing robust estimates of gestational age in SGA infants.

We have included mention of this in the discussion:

Discussion: "Although limited sample size prevented evaluation of the model's amongst small-for gestational age or low-birthweight infants, recent work from our group has demonstrated comparable accuracy amongst this infant subpopulation¹⁴. This is particularly important when considering the potential to implement metabolic dating tools in LMIC given the prevalence of low birthweight infants in low-resource communities."

7. It is stated in the introduction that alternative methods of GA estimation are being sought. Alternative methods are not discussed anywhere in this research paper - foot lengths, cranial ultrasound, etc. Alternative methods' accuracy are also not compared to the current metabolic models.

Author Response: Thank you. The originally drafted version of this manuscript was intended to be a brief communication piece, and we now appreciate the opportunity to expand on the content of the paper. We should clarify that the 'alternative methods' to which we speak in the introduction refer to alternative postnatal methods, as the key issue in LMIC remains access to antenatal care services, with women often seeking health care very close to the time of delivery. This has been clarified in the introduction:

Introduction: "In jurisdictions with challenging socioeconomic conditions and/or limited access to antenatal care (and thus ultrasound technology) due to rurality, determination of gestational age can be challenging. Other antenatal dating methods, including last menstrual period and fundal height measurements, are hampered by poor recall history and a high prevalence of growth restriction. Where prenatal estimations are unavailable or unreliable, a variety of standardized fetal assessments have been developed for clinicians seeking to determine fetal maturation after birth. Commonly used postnatal assessments that score infants on neurologic and physical criteria are only accurate to within three to four weeks of true, ultrasound-validated gestational age³. World health and philanthropic organizations are now seeking novel ways of determining gestational age at the time of birth, both to improve individual care and to provide reliable, high-quality data for population surveillance."

Comparison of the precision of our models against other more commonly used postnatal dating measures (including the Dubowitz or Ballard Scores) are now more clearly addressed in the introduction (above) and in the Discussion (below):

Discussion: "Precision of metabolic dating methods to within 1-2 weeks compares favourably to other commonly used postnatal dating methods, the accuracy of which vary widely (3-4 weeks) depending on the method, level of training of the specialist performing the measurements and if the child is small for gestational age."

8. The cost of metabolic screening is excessive in most non-American/ European countries and often not even available in lower/ middle income countries. This aspect has not been mentioned in the study - it is thus difficult to accept the clinical applicability of this model in these countries.

Author Response: We thank the reviewer, as this is indeed an important observation. Expanded newborn screening is not accessible to many individual families, nor is it available in some LMIC. Our revisions clarify that the greatest potential benefits of metabolic dating are likely to be for provision of reliable population estimates of preterm birth (ie population surveillance) and we highlight the need to streamline models such that they may be feasibly implemented, without sacrificing accuracy.

Discussion: "This is particularly important when considering the potential to implement metabolic dating tools in LMIC given the prevalence of low birthweight infants in low-resource communities. In addition to continuous estimates, provision of gestational age estimates across dichotomous thresholds (eg ≥ 37 or < 37 weeks gestational age) may be useful for the purpose of population surveillance. Second, currently published models are complex, and require data on a large number of analytes measured by mass spectrometry. Ultimately, simplification of models to reduce the number metabolic variables while maintaining model performance will be required to streamline the approach for scalable, cost-effective, implementation."

Responses to Reviewer 3 Comments:

1. "The MSE is the sum of each of those residual errors" -> please replace the word "sum" by "mean".

Author Response: We thank the reviewer for her careful review of the manuscript. The above noted change has been made.

2. An "RMSE of 1 may be interpreted as the regression model estimating gestational age within 1 week on average": This statement is confusing. I recommend removing it (in the methods section and whenever it occurs in the Results)

Author Response: Thank you – we have removed this potentially confusing phrase, and similar mentions from the manuscript.

3. Typo: "Missing data occurred for a very proportion": a word is missing.

Author Response: Thank you – we have updated the text: "Missing data occurred for a very small proportion..."